# Epstein–Barr Virus Infection of Pseudostratified Nasopharyngeal Epithelium Disrupts Epithelial Integrity

**DOI:** 10.3390/cancers12092722

**Published:** 2020-09-22

**Authors:** Fenggang Yu, Yanan Lu, Yingying Li, Yuji Uchio, Utomo Andi Pangnguriseng, Andy Visi Kartika, Hisashi Iizasa, Hironori Yoshiyama, Kwok Seng Loh

**Affiliations:** 1Department of Otolaryngology—Head and Neck Surgery, Yong Loo Lin School of Medicine, National University of Singapore, Singapore 119228, Singapore; m.s.yanan@gmail.com (Y.L.); entlks@nus.edu.sg (K.S.L.); 2Institute of Life Science, Yinfeng Biological Group, Jinan 250000, China; 3Department of Biomedical Engineering, National University of Singapore, Singapore 117583, Singapore; muzishuiyu@gmail.com; 4Department of Orthopaedic Surgery, Faculty of Medicine, Shimane University, 89-1 Enya, Izumo, Shimane 693-8504, Japan; uchio@med.shimane-u.ac.jp (Y.U.); utomo.andipangnguriseng@umi.ac.id (U.A.P.); 5Department of Microbiology, Faculty of Medicine, Shimane University, 89-1 Enya, Izumo, Shimane 693-8504, Japan; vkartika@med.shimane-u.ac.jp (A.V.K.); iizasah@med.shimane-u.ac.jp (H.I.); 6Department of Pathology Anatomy, Faculty of Medicine, University of Muslim Indonesia, Jl. Urip Sumoharjo KM.5, Makassar, Sulawesi 90231, Indonesia; 7Department of Otolaryngology—Head and Neck Surgery, National University Health System, 1E Kent Ridge Rd, Singapore 119228, Singapore

**Keywords:** Epstein–Barr virus, nasopharyngeal cancer, pseudostratified epithelium, in vitro EBV infection model

## Abstract

**Simple Summary:**

Nasopharyngeal carcinoma is associated with Epstein-Barr virus (EBV) infection and originates junction of the oropharynx and nasal cavity, where stratified squamous epithelium and respiratory epithelium are the lining. To elucidate the mechanisms by which EBV transforms the nasopharyngeal epithelium, a pseudostratified multiple-layer model with cilia forming on the apical surface by air-liquid interface (ALI) culture of primary nasopharyngeal epithelial cells was established. We showed: (1) ALI cultures formed stratified epithelia and maintained the diversity of cells found in the airway epithelium, such as ciliated, muco-secretory, and basal cells. (2) Polarized stratified epithelium was more susceptible to EBV infection than monolayer cells. (3) EBV infection in ALI cultures was verified by showing EBV-encoded RNA expressions. (4) EBV infection disrupted the integrity of the epithelium. Thus, our model can be used not only to examine the pathogenesis of pre-neoplastic EBV-infected cells, but also to develop anti-EBV therapy or early stage NPC treatment.

**Abstract:**

Epstein–Barr virus (EBV) is a human oncogenic virus that causes several types of tumor, such as Burkitt’s lymphoma and nasopharyngeal carcinoma (NPC). NPC tumor cells are clonal expansions of latently EBV-infected epithelial cells. However, the mechanisms by which EBV transforms the nasopharyngeal epithelium is hampered, because of the lack of good in vitro model to pursue oncogenic process. Our primary nasopharyngeal epithelial cell cultures developed pseudostratified epithelium at the air-liquid interface, which was susceptible to EBV infection. Using the highly sensitive RNA in situ hybridization technique, we detected viral infection in diverse cell types, including ciliated cells, goblet cells, and basal cells. EBV-encoded small RNA-positive cells were more frequently detected in the suprabasal layer than in the basal layer. We established the most physiologically relevant EBV infection model of nasopharyngeal epithelial cells. This model will advance our understanding of EBV pathogenesis in the development of NPC.

## 1. Introduction

Epstein–Barr virus (EBV) is a ubiquitous gamma herpesvirus that infects both B-lymphocytes and epithelial cells [1]. The EBV infection of these cells can lead to the clonal expansion of a persistently infected cell, and sometimes induces cancer, such as nasopharyngeal carcinoma (NPC) [2,3]. However, the mechanisms by which EBV transforms the nasopharyngeal epithelium remain mostly unknown. To further understand these mechanisms, we need to develop more physiologically relevant in vitro epithelial infection models.

A variety of in vitro epithelial cell models for EBV infection have been developed, ranging from monolayer unpolarized epithelial cells to polarized epithelial cells, using oropharyngeal or tonsillar epithelial cells [4,5,6,7,8]. Although many useful data have been generated using these models, they may not closely represent the physiological condition in the nasopharynx, for the following reasons. First, most epithelial cells were derived from oropharyngeal or tonsillar epithelium, but cells from the nasopharynx were scarcely used, because they were difficult to obtain. Second, the nasopharynx is located at the junction of the oropharynx and nasal cavity, where stratified squamous epithelium and respiratory epithelium are the lining. The stratified squamous epithelium is localized on the inferior anterior, inferior posterior and anterior lateral walls, while the respiratory epithelium is localized around the nasal choanae and roof of the posterior wall. Lastly, abundant lymphoid tissues are found underneath the nasopharynx, which is structurally distinct from other anatomical regions, such as oropharyngeal or tonsillar epithelia.

In this study, we established a conventional monolayer model and a pseudostratified multiple-layer model, using epithelial cells derived from rare and precious nasopharynx specimens, to mimic the squamous and respiratory parts of the nasopharyngeal epithelium. We investigated EBV infection using these models (Figure 1a).

## 2. Results

### 2.1. Establishment of Primary Nasopharyngeal Epithelial Cell Cultures

Cells derived from normal nasopharynx biopsies could grow directly on plastic wells without feeder cells. Cells exhibited a typical polygonal epithelial appearance, and strongly expressed epithelial cell markers: pan-cytokeratin (pan-CK) and basal cell marker p63 (Figure 1b). The pseudostratified epithelium was formed in a transwell insert after 3 to 4 weeks of air–liquid interface (ALI) culturing. A movie of ciliated beating cells of the pseudostratified epithelium is presented (Appendix A). Hematoxylin and eosin (H&E) staining of a longitudinal cryosection showed that the epithelium was composed of multiple layers, with cilia on the apical surface (Figure 1b).

### 2.2. EBV Infects a Conventional Monolayer of Nasopharyngeal Epithelial Cells

Our previous study indicated that EBV transfer of nasopharyngeal epithelial cells is more efficiently mediated by cell-to-cell contact than by cell-free infection [9]. Therefore, instead of using cell-free infection, we employed cell-mediated infection by adding virus-producing Akata cells infected with recombinant EBV, containing the green fluorescent protein (GFP) gene at the viral BXLF1 locus (Akata GFP-EBV cells) [10], onto monolayer nasopharyngeal epithelial cells. Because Akata GFP-EBV cells produce viruses upon stimulation with anti-human immunoglobulin G (IgG) [11], EBV-infected epithelial cells expressed GFP. As a result, a conventional monolayer of nasopharyngeal epithelial cells was susceptible to EBV infection, in spite of a low efficiency of 0.26%, based on the quantification of GFP positive infected epithelial cells (Figure 1c).

### 2.3. EBV Infects Pseudostratified Epithelia Formed by Primary Nasopharyngeal Epithelial Cells

Due to the fact that ALI cultures were generated from primary nasopharyngeal epithelial cells and reflect in vivo epithelial tissue more accurately than monolayer cultures, we expected that EBV would infect ALI cultures more efficiently than monolayer cultures. The differentiation state of day 28 cultures was assessed by transepithelial electrical resistance (TEER) and ciliary beat frequency (CBF) measurements, and only fully stratified epithelium with an average TEER above 4000 Ω × m^2^ and CBF > 7 Hz were used for EBV infection. However, direct detection of GFP expression on EBV infected cells was difficult, even after the signals were enhanced by GFP antibody staining. 

On the other hand, we could confirm EBV infection by RNAscope in situ hybridization (ISH), because the assay was sensitive enough to detect at single transcript level. The sensitivity and specificity of RNAscope ISH were validated by hybridizing EBV positive cell line or NPC tissue with an EBV-encoded small RNA1 (EBER1) probe, because EBERs (EBER1 and EBER2) are strongly expressed in EBV infected cells, and serve as sensitive targets for detecting EBV infection (Figure 2) [12]. We used C666-1 cells for EBER1 detection, because C666-1 cell is a representative NPC cell line that expressed EBERs, EBNA1, *Bam*HI-A right transcripts, latent membrane protein 1 (LMP1), and LMP 2A/B [13]. EBER1-staining of C666-1 cells showed both a diffuse pattern and a discrete speckled pattern (Figure 2b). Similarly, cytokeratin positive NPC tumor cells showed both a diffuse staining pattern and a discrete speckled staining pattern (Figure 2d).

We then examined whether EBV infects pseudostratified epithelia were formed by primary nasopharyngeal epithelial cells. Five days post-infection (d.p.i.), we performed hybridization with an EBER1 probe that detects EBV infections, and a BRLF1 probe that detects lytic infections (Figure 2 and Figure 3). All cell types, including basal cells (p63-positive) (Figure 3a,b), goblet cells (MUC5AC-positive) (Figure 3c,d), and ciliated cells (βIV-tubulin-positive) (Figure 3e,f) were able to confirm EBV infection, by showing positive signals for EBER1 or BRLF1. In this system, only few goblet cells with strong MUC5AC staining on the apical side of the cells were detected (Figure 3c,d). There were significantly more EBV signals in the suprabasal layer, mainly composed of βIV-tubulin-positive ciliated cells, than the basal layer (Figure 3e,f).

Notably, most of the EBER-positive signals were localized in the nucleus, but not elsewhere in the cytoplasm or cilia (Figure 3a,c,e). Substantial numbers of EBER-positive signals could also be observed in EBV-positive NPC tumor cells (Appendix A).

### 2.4. Predominant EBV Infection of the Suprabasal Layer Cells

Either EBER1 or BRLF1 signals were counted in more than 200 EBV-infected nasopharyngeal epithelial cells (Figure 4). EBER positive cells are 65% with 4.5 puncta/cell and 25.5% with 2.6 puncta/cell in suprabasal layer cells and basal layer cells, respectively (Figure 4a,c). BRLF1 positive cells are 35.5% with 4.0 puncta/cell and 28.3% with 2.0 puncta/cell in suprabasal layer cells and basal layer cells, respectively (Figure 4b,d). EBER positive cells were more frequently observed in the suprabasal layer than in the basal layer (*p* < 0.05, Figure 4a,c). 

### 2.5. EBV Infection Disrupts the Integrity of the Epithelium

Cilia beating plays a key role in airway-defense. CBF was initially reduced by the addition of EBV-infected cells and increased after removing these cells (Figure 5a). The TEER value provides information about the uniformity of the multi-layer cells and the integrity of the tight junctions between the polarized cells. When infected cells were added to the transwell, the TEER dropped sharply. On the other hand, TEER did not decrease with the addition of control cells (Figure 5b). The integrity of the epithelium was disrupted by EBV infection, but not by the addition of cells.

## 3. Discussion

The association between EBV and epithelial malignancies, including NPC, has been known for decades. However, the life cycle of EBV during carcinogenesis is not well known, because of the lack of good in vitro models. Here, we successfully established an in vitro model of EBV infection using primary nasopharyngeal epithelial cells. The stratified multi-layer nasopharyngeal epithelial cells with several types of differentiated cells (Figure 1b (iv)) were infected with EBV (Figure 1c).

Our model is physiologically relevant because of the following aspects. Firstly, we use authentic epithelial cells isolated from the nasopharynx where NPC originates. Secondly, two mechanisms of EBV infection is thought to happen in human body, one is cell-free infection mediated by saliva and the other is cell-to-cell infection mediated by EBV-infected or EBV-loaded B lymphocytes [14,15]. We showed that cell-to-cell infection is much more efficient than cell-free infection in the experimental animal model [9]. B lymphocyte-mediated infection must be important for epithelial EBV infection, because absence of a detectable latent EBV infection in persons with X-linked agammaglobulinemia who lack mature B lymphocytes [16]. Finally, in our previous report, 3 of 10 samples showed less than 10 EBER puncta/infected tumor cells [17]. The range of infection was similar to the current model (Appendix A).

A relevant study by Temple et al. demonstrated in vitro EBV infection using primary oral keratinocytes raft cultures, polarized but not completely stratified cells. They showed that EBV infection resulted in viral production and spread throughout the culture, without establishing latent infection [18]. We showed that polarized stratified epithelium is more susceptible to EBV infection than monolayer cells (Figure 1c and Figure 3). Moreover, we detected both latent and lytic infection signals, as evidenced by EBER1-positive and BRLF1-positive cells (Figure 3a,c,e). The high sensitivity and specificity of our probes have been validated in a dozen clinical samples [17]. In addition, because EBER expression is the gold standard to prove EBV infection, we established a model of EBV infection in nasopharyngeal epithelial cells.

Temple et al. used raft cultures of primary oral keratinocytes [18], that are polarized but not completely stratified cells. In contrast, our ALI cultures of nasopharyngeal epithelial cells form stratified epithelia and maintain the diversity of cells found in the airway epithelium, such as ciliated, muco-secretory, and basal cells (Figure 3, Appendix A). 

Another relevant ALI culture by Caves et al. used HK1 cells derived from immortalized NPC cells, but not primary cells [8]. We and Caves et al. showed that a differentiation of latently EBV-infected cells triggered viral lytic replication. However, Caves et al. are different from ours in several ways. First, the HK1 cell was derived from a squamous carcinoma of the nasopharynx and was negative for EBV [19]. Second, the HK1 cells were infected with EBV by Akata GFP-EBV cells prior to placing on the transwell. Third, Caves’ model was a polarized model, but not a real stratified model.

A high number of EBER1 and BRLF1 puncta were mostly observed in suprabasal layer (Figure 4). A cellular transcription factor, KLF4 was shown to induce differentiation-dependent lytic EBV infection in the mix culture of normal oral keratinocyte and Akata cells [20]. Though we did not investigate the detailed mechanisms involved, viral lytic reactivation could be induced by such kinds of cellular transcription factors. 

Although we did not check the cytopathic effects at the individual cell level, we observed that EBV infection disrupted the integrity of the epithelium (Figure 5b). The cilia beating was temporarily compromised, but was eventually restored (Figure 5a). The disruption of epithelial integrity in EBV-infected cells is reflected in numerous reports, that show increased motility or anchorage independence in EBV-infected gastric, nasopharyngeal, and squamous cells [5,21,22]. Nevertheless, we need to determine whether our observation that EBV infection disrupts the epithelial integrity is unaffected by cytokines/exosomes released from activated Akata cells, by expanding the next experimental size.

Because EBV genomes in NPC tumor cells are monoclonal, tumor cells are clonal expansions of latently EBV infected nasopharyngeal cells [23]. However, the effort to elucidate mechanisms by which EBV transforms nasopharyngeal epithelium are hampered by the lack of in vitro transformation model. The current understanding is that low-grade preinvasive lesions become susceptible to latent EBV infection, because EBV can be detected in precancerous, but rarely in normal, pharyngeal epithelium [24,25]. We have previously established the culturing system of cells derived from normal nasopharynx of NPC patients [26]. Here, we showed that noncancerous stratified pharyngeal epithelia were susceptible to EBV infection, and their differentiation leads to lytic replication. Our new finding that EBV infection disrupted epithelial integrity may predict the carcinogenic process. ALI infection model utilizing either precancerous epithelial cells or epithelial cells that have received transforming stimuli could show a difference not only in EBV susceptibility, but also in epithelial integrity.

Taken together, we established the most relevant EBV infection model of stratified nasopharyngeal epithelia. This model can be used not only to examine the pathogenesis of pre-neoplastic EBV-infected cells, but also to develop anti-EBV therapy or early stage NPC treatment.

## 4. Materials and Methods

### 4.1. Biopsy Collection and Cell Culture

The present study was approved by the National Healthcare Group Domain-Specific Review Board of Singapore (Reference No.: DSRB-B/10/337). Informed and written consent was obtained from all patients. All procedures were performed in accordance with the approved guidelines. The specimen was washed extensively in Hank’s balanced salt solution, containing penicillin, streptomycin, and amphotericin B. The specimen was digested in 10 mg/mL of Dispase II (Sigma-Aldrich, St. Louis, MO, USA) at 4 °C overnight, followed by dissociation through repetitive pipetting. The digestion was stopped by adding DMEM (Thermo Fisher Scientific, Waltham, MA, USA), containing 10% fetal calf serum (FCS) (Thermo Fisher Scientific). The dissociated cells were washed twice and cultured as a monolayer in DMEM/F12 (Thermo Fisher Scientific), containing 10 ng/mL of human epithelial growth factor (Thermo Fisher Scientific), 5 μg/mL of insulin (Sigma-Aldrich), 0.1 nM of cholera toxin (Sigma-Aldrich), 0.5 μg/mL of hydrocortisone (Sigma-Aldrich), 2 nM/mL of 3,3’,5-triiodo-l-thyronine (T3) (Sigma-Aldrich), 10 μL/mL of N2 supplement (Thermo Fisher Scientific) and 100 IU/mL of antibiotic–antimycotic (Thermo Fisher Scientific). Epithelial cell identity was confirmed by immunofluorescent staining of cytokeratin (AE1/AE3, Abcam, Cambridge, UK) and p63 (A4A, Abcam) as previously described [26,27]. 

HONE1 cells [28] and C666-1 cells [13] were maintained in RPMI1640 (Sigma-Aldrich) containing 10% FCS.

### 4.2. ALI Culture

We resuspended 1 × 10^5^ primary cultured EBV negative cells in 100 μL of B-ALI^TM^ growth medium (Lonza, Walkersville, MD, USA) and pipetted it onto 24-well transwell inserts (0.4 μm pore size, Thermo Fisher Scientific), with 400 μL of B-ALI^TM^ growth medium, that was added to the basal chamber. On day 3 after seeding, the B-ALI^TM^ growth medium from the apical and basal chambers was removed, and 400 μL of B-ALI^TM^ differentiation medium was added to the basal chamber only. The differentiation medium of the basal chamber was changed every other day during the 4-week culture period [19,20,21]. The ALI cultures were observed with an inverted microscope (Olympus, Tokyo, Japan) at 400X magnification. TEER and CBF were measured [18].

### 4.3. TEER and CBF Measurements

CBF was recorded and analyzed automatically by using the Sisson-Ammons Video Analysis system (SAVA, Ammons Engineering, Clio, MI, USA) [29,30].

TEER was measured with EVOM2 instrument (World Precision Instruments, Sarasota, FL, USA). Before measurement, electrodes were equilibrated and sterilized according to the manufacturer’s recommendations. Two hundred microliters of culture medium were added in the upper compartment of the cell culture system. The Ohmic resistance of a blank (culture insert without cells) was measured in parallel. To obtain the sample resistance, the blank value was subtracted from the total resistance of the sample.

### 4.4. Immunostaining and ISH

Cell types were determined by staining cryosections of the transwells with indicated lineage markers: β IV tubulin (ONS.1A6, Abcam) for ciliated cells, MUC5AC (2–11M1, Abcam) for goblet cells, and p63 (A4A, Abcam) for basal cells. Alexa Fluor 594 conjugated goat anti-mouse IgG was used as secondary antibody (Thermo Fisher Scientific). EBER1 and BRLF1 probes were purchased from Advanced Cell Diagnostics (Newark, CA, USA). The EBER1 probe (#310271) was designed based on human herpesvirus 4 (HHV4) isolate SDTW400 EBER1 and EBER2 gene sequence (Accession Number: KP195701.1). The probe consists of 2 ZZ pairs, spanning nucleotide positions 39 to 120. The BRLF1 probe (#450401) was designed according to coding sequence of BRLF1 of HHV4 complete genome (Accession Number: NC_007605.1). The probe consisted of 20 ZZ pairs, spanning nucleotide positions 2 to 1557. In combination with immunohistochemistry, a type of ISH, RNAscope, was performed as previously described [9,15]. Formalin-fixed, paraffin-embedded transwell sections and NPC tissues (4 μm) were deparaffinized, and were then also stained using a standard H&E staining protocol [31].

### 4.5. Virus Production and Infection

Two Burkitt’s lymphoma cell lines, one is EBV-negative Ramos cells and the other is GFP-tagged EBV infected Akata cells [10], were maintained in RPMI 1640 (Sigma-Aldrich), supplemented with 10% FBS. Akata cells were stimulated by 100 µg/mL of rabbit IgG-Fab fraction against human IgG (Jackson Immunoresearch, West Grove, PA, USA) for 24 h, to induce lytic EBV production [11]. Cell suspensions were centrifuged for 5 min at 300× *g* and washed once with phosphate buffered saline (PBS). The resultant Akata cells were resuspended in PBS to a concentration of 0.25 × 10^6^ cells/100 µL, and then 200 µL was added to each well (24 well plate) of monolayer cells or 200 μL to the top chamber of the transwell (Figure 1a). EBV-negative Ramos cells were similarly treated with rabbit anti-human IgG-Fab and added into wells as a sham control.

### 4.6. Statistical Analysis

A Mann–Whitney test was used to analyze and to test whether there was a difference between two independent groups, which was expressed as mean ± standard deviation (SD). All experiments were repeated three or five times. A probability (*p*) value < 0.05 was considered statistically significant.

## 5. Conclusions

The present study showed that noncancerous stratified pharyngeal epithelia generated by ALI culture were susceptible to EBV infection, and cell differentiation leads to lytic viral replication. Moreover, EBV infection disrupted the integrity of the epithelium that may increase cell motility. The ALI infection model is useful for investigating the tumorigenic process of NPC and for developing anti-EBV chemotherapeutic agents to treat NPC patients.

## Figures and Tables

**Figure 1 cancers-12-02722-f001:**
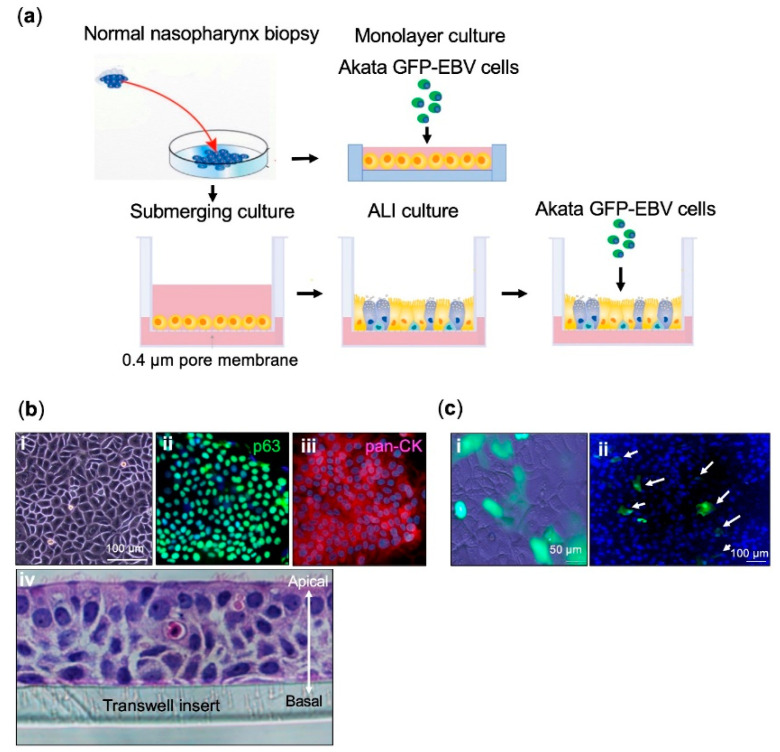
An in vitro Epstein–Barr virus (EBV) infection model of pseudostratified epithelium derived from primary nasopharyngeal epithelial cell cultures: (**a**) diagram of the experimental procedure. Green fluorescent protein (GFP)-tagged virus-producing Akata cells were added into monolayer culture or air–liquid interface (ALI) culture of nasopharyngeal epithelial cells for infection. (**b**) Morphology and marker expression in nasopharyngeal epithelial cell cultures. Representative microscopic images depicting cells in monolayer (**i**) and cells expressing pan-CK and p63 determined by immunofluorescence (**ii**,**iii**). Hematoxylin and eosin staining of the longitudinal section of day 28 ALI culture (**iv**). It consists of multi-layer of cells with cilia forming on the apical surface. (**c**) In vitro EBV infection of the monolayer of nasopharyngeal epithelial cells. Merged picture of phase contrast and GFP channel (**i**). Merged picture of GFP and DAPI (nuclear counterstain in blue) (**ii**). Infection efficiency was determined by calculating GFP positive cells over total cells in DAPI.

**Figure 2 cancers-12-02722-f002:**
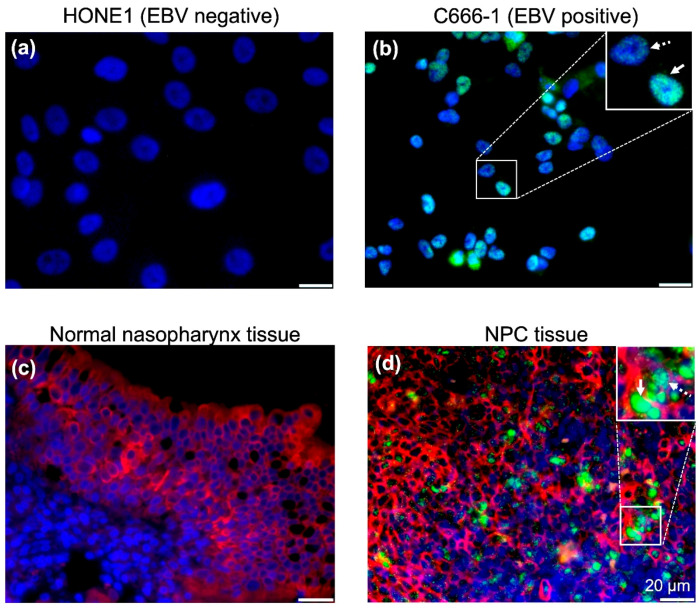
Detection of EBER1 signals by RNAscope in situ hybridization system. EBER in situ hybridization (ISH) was performed on EBV-positive and EBV-negative cells using the EBER1 probe (#310271) by RNAscope detection system. (**a**) HONE1 cell, (**b**) C666-1 cell, (**c**) normal pharynx tissue, and (**d**) NPC tumor tissue were examined to detect EBER1, respectively. Solid arrow that shows diffuse staining pattern as well as dotted arrow that indicates discrete speckled staining pattern are indicated in the inserts. Nuclei are stained blue by DAPI. EBER1 ISH signals are shown in green. Cytokeratin immunostaining signals are shown in red.

**Figure 3 cancers-12-02722-f003:**
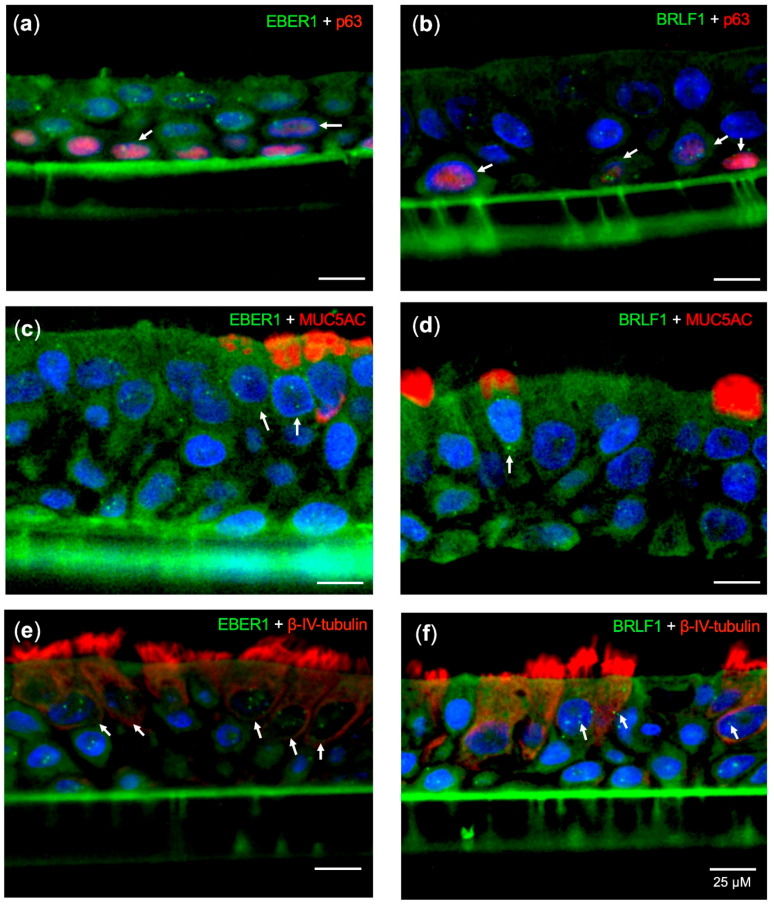
EBV infection of all cell types in the ALI culture. Longitudinal sections of EBV-infected transwells were assayed by immunohistochemistry of cell lineage markers and EBV-specific probes. (**a**) EBER1 and p63, (**b**) BRLF1 and p63, (**c**) EBER1 and MUC5AC, (**d**) BRLF1 and MUC5AC, (**e**) EBER1 and βIV-tubulin, and (**f**) BRLF1 and βIV-tubulin were stained, respectively. ISH signals of strong, discrete puncta in the nuclei are shown in green, and immunostaining signals are shown in red. White arrows indicate cells containing both green and red signals.

**Figure 4 cancers-12-02722-f004:**
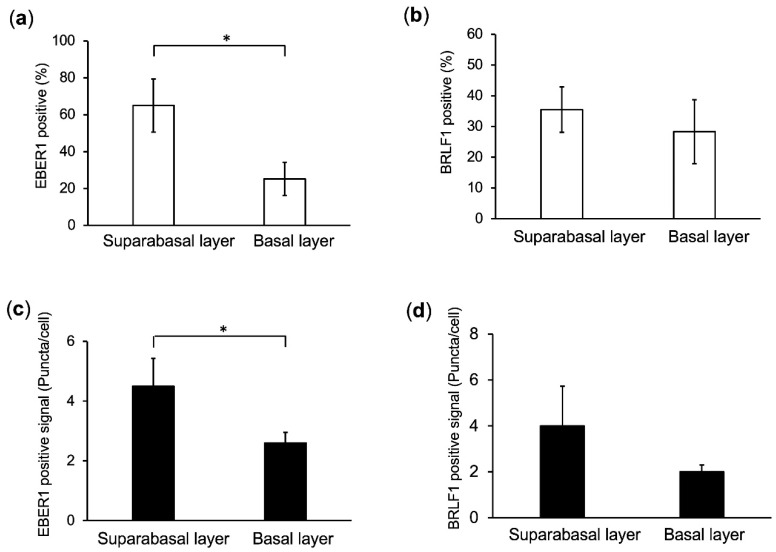
EBV infection is predominantly located in the suprabasal layer. ISH signal (EBER1 or BRLF1 mRNA) and p63 are stained in green and red, respectively. (**a**,**b**) Percentage of EBER1 positive (**a**) or BRLF1 positive (**b**) cells in suprabasal and basal layers are indicated by white columns, respectively. (**c**,**d**) Number of puncta showing EBER1 (**c**) or BRLF1 (**d**) ISH signals per cell are indicated by black columns, respectively. *: *p* < 0.05.

**Figure 5 cancers-12-02722-f005:**
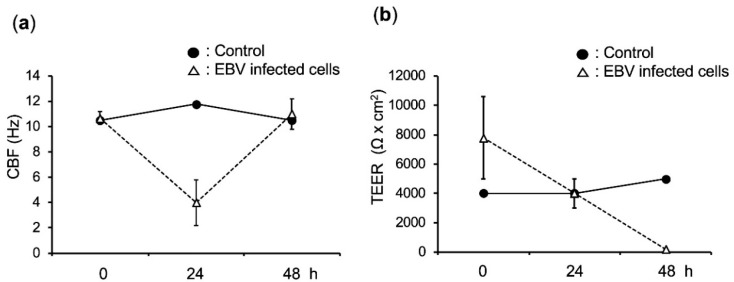
Effects of EBV infection on CBF and TEER. (**a**) CBF and (**b**) TEER were measured at different time points (0, 24, and 48 h). The virus-producing Akata cells were added for infection and removed after 24 h by washing. Control cells were EBV-negative Ramos cells.

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
