# Peer review of "Epstein–Barr Virus Infection of Pseudostratified Nasopharyngeal Epithelium Disrupts Epithelial Integrity"

_cancers, 2020, doi:10.3390/cancers12092722_

Round 1

Reviewer 1 Report

The article „Epstein-Barr virus infection of pseudostratified 3 nasopharyngeal epithelium disrupts epithelial integrity“ represents a nice contribution for EBV infections pathophysiology. The idea of testing in vitro model is interesting, and the method of infection of the cells is known and widely used. I believe, that, the idea of disruption of the epithelium and facilitation of infections spread is correct. It is known for example, the risk of malignancy in smokers is 10 times higher when drinking strong alcoholic drinks. The alcohol melts polycyclic aromatic hydrocarbons and facilitates spread to the deeper epithelial layers. From the clinical point of view the idea is unique.

The article is well written, readable, and the amount of illustrations covers the readers demands.

I consider this rewritten version of the article interesting for readers.

Author Response

Thank you very much for your encouraging comments. We hope our work inspires more people in this research field.

Reviewer 2 Report

Reviewer Comments:

In this manuscript, Yu et al. have set out to establish a more physiologically relevant pseudostratified multi-layer tissue model system using the normal nasopharyngeal epithelial cells. They then investigated EBV infection in this model by co-culturing the anti-IgG treated EBV+ Akata cells. Based on the EBER or BRLF1 staining, they concluded that all the analyzed cell types were infected with EBV and the infection mainly occurred in the suprabasal layer. Lastly, based on the TEER readings, they concluded that EBV infection disrupted tight junctions between polarized cells and the integrity of the epithelium.  Overall, the authors reported an interesting model system which may benefit the EBV research field. However, there are several concerns raised by this reviewer.   

Major issues:

(1) The EBER staining data are not convincing.  EBER is the most abundant transcripts in EBV+ cells and usually show diffused staining pattern.  However, in Figure 2, EBER shows an unusual discrete speckled staining pattern, suggesting a possible artifact.  

(2) The authors stated that "In addition, because EBER expression is the gold standard to prove EBV infection, they established a model of latent infection in nasopharyngeal epithelial cells”.  This is not accurate.  The conclusion of latent infection is not sufficiently supported by the data provided.  EBERs are expressed in both latent and lytic viral infected cells.  The authors need to examine the true viral latent markers to determine if a latency can be established and what type of latency can be established.   

(3) The authors concluded that EBV infection disrupts the integrity of the epithelium. This is arguably the most important discovery reported by the study.  However, the experiment was not well controlled.  The authors should use EBVneg Akata cells or a BL cell line carrying the IgG-BCR.  The Ramos cells have IgM-BCR on its surface. The anti-IgG treatment won’t induce Ramos activation.  A proper control is important since it’s likely that the activated Akata cells can release cytokines/exosomes to affect the epithelium integrity. 

(4) The study is pretty descriptive and no mechanism insight’s provided.

Minor issues:

(1) Supplemental Figure 1 is missing.

Author Response

Major issues:

  • The EBER staining data are not convincing. EBER is the most abundant transcripts in EBV+ cells and usually show diffused staining pattern.  However, in Figure 2, EBER shows an unusual discrete speckled staining pattern, suggesting a possible artifact.

→The sensitivity and specificity of our RNAscope probes have been validated in both C666-1 and NPC tissues with proper controls in the new Figure 2. The in situ hybridization protocols have been established in our lab and have been reported (Yu F et al. Presence of lytic Epstein-Barr virus infection in nasopharyngeal carcinoma. Head Neck 2018, 40, 1515–23. [Ref. 17]). In the new Figure 2, C666-1 NPC cell line showed not only diffuse staining pattern, but also discrete speckled staining pattern (Fig. 2b). The discrete speckled staining pattern is more frequently observed in NPC tissues (Fig. 2d). The patterns are depending on the amount of viral RNA transcripts. The results were described from line 103 to line 111 in the revised manuscript.

  • The authors stated that "In addition, because EBER expression is the gold standard to prove EBV infection, they established a model of latent infection in nasopharyngeal epithelial cells”.  This is not accurate.  The conclusion of latent infection is not sufficiently supported by the data provided.  EBERs are expressed in both latent and lytic viral infected cells.  The authors need to examine the true viral latent markers to determine if a latency can be established and what type of latency can be established.

→ Thank you very much for important suggestions. We agree that EBER is expressed not only in latencies 0, I, II, and III, but also in lytic infection. On the other hand, it is reported that the level of EBER expression is most abundant in latent infection. Although their half lives are seemingly long, EBERs (EBER1 and EBER2) are downregulated on the transcriptional level during the lytic viral replication (Greifenegger N, Jäger M, Kunz-Schughart LA, Wolf H, Schwarzmann F. Epstein-Barr virus small RNA (EBER) genes: differential regulation during lytic viral replication. J Virol. 1998, 72, 9323-9328.). Considering this situation and in order to describe accurately, the words ‘model of latent infection’ is replaced with ‘model of EBV infection’ on line 187 of the revised manuscript.

  • The authors concluded that EBV infection disrupts the integrity of the epithelium. This is arguably the most important discovery reported by the study.  However, the experiment was not well controlled.  The authors should use EBVneg Akata cells or a BL cell line carrying the IgG-BCR.  The Ramos cells have IgM-BCR on its surface. The anti-IgG treatment won’t induce Ramos activation.  A proper control is important since it’s likely that the activated Akata cells can release cytokines/exosomes to affect the epithelium integrity. 

→ Thank you very much for the helpful suggestion. We used Ramos cells that were not activated by anti-IgG antibody. We understand that it is better to use EBV negative Akata cells to evaluate effect of EBV on disruption of epithelial integrity. For the foreseeable future, it is very difficult to do more experiments, because we could not obtain human nasopharyngeal samples due to COVID-19 issue. Instead, we conducted a literature survey to support our observation and made additional discussions from line 206 to line 211 in the revised manuscript.

  • The study is pretty descriptive and no mechanism insight’s provided.

→ We totally understand the reviewer’s opinions. This short report mainly describes the establishment of the model of EBV infection of multilayer of nasopharyngeal pseudostratified epithelium. We have faced tremendous difficulties in detecting EBV infected cells by conventional methods, such as direct GFP fluorescence, EBNA1 antibody staining and conventional EBER-ISH etc. As an initial study, we have barely touched molecular mechanisms. After developing a more frequent infection model, we will definitely investigate mechanisms underlying EBV infection in nasopharyngeal epithelium.

Minor issues:

  • Supplemental Figure 1 is missing.

→ We apologize for the mistake. We have wrongly labeled supplemental data in the last submission. In the current revised manuscript, we have ‘Supplemental video 1’ and ‘Supplemental figure 1’. We confirmed the accuracy for each label and its description in the revision.

Reviewer 3 Report

Fenggang Yu et al. focused on the description of mechanisms by which Epstein-Barr virus transforms the nasopharyngeal epithelium. Pseudostratified epithelia (PSE) are widespread and diverse tissue arrangements, and many PSE are organ precursors in a variety of organisms. While cells in PSE, like other epithelial cells, feature apico-basal polarity, they generally are more elongated and their nuclei are more densely packed within the tissue. In addition, nuclei in PSE undergo interkinetic nuclear migration (IKNM, also referred to as INM), whereby all mitotic events occur at the apical surface of the elongated epithelium.The authors, according to my knowledge, for the first time, established a PSE multiple layer model, using epithelial cells derived from nasopharynx specimens to mimic the squamous and respiratory parts of the nasopharyngeal epithelium. Fenggang Yu et al. investigated EBV infection using this model. Their previous study, entitled "Non-malignant epithelial cells preferentially proliferate from nasopharyngeal carcinoma biopsy cultured under conditionally reprogrammed conditions", published in 2017, was an excellent introduction to the present study. New model, highly similar to the physiological conditions, seems to be useful not only to examine the pathogenesis of pre-neoplastic EBV-infected cells, but also to develop anti-EBV therapy or early stage nasopharyngeal cancer treatment.
Summing up, I believe that the work, due to its novelty, should be published. The methodology has been precisely described and can be repeated in any specialist immunology laboratory. The authors, in their subsequent experiments, should evaluate the impact of EBV and human papilloma virus (HPV) co-infection on PSE.

Author Response

Thank you very much for valuable comments. We also expect our model can be applied to co-infection of EBV and human papilloma virus (HPV) on PSE. It is well worth exploring this novel idea.

This manuscript is a resubmission of an earlier submission. The following is a list of the peer review reports and author responses from that submission.

Round 1

Reviewer 1 Report

The authors are focused on the cell models of Epstein Barr virus effects on epithelium of the nasopharynx. They showed enormous scientific effort and diligent work to elucidate the effect of the virus on epithelium. The title grabs the attention immediately and the abstract appetizes interest. The different types of cells used in model are exceptional. The figures and graphs are sufficient to draw the image of the research. The only one think which in my opinion should be improved is the discussion. The references are old, I believe, that the readers would appreciate more up to date information regarding this particular and narrow scientific field. I miss the possible importance of the research for the diagnosis, patient and possible treatment.

Author Response

Thank you very much for valuable comments. According to the suggestions, we changed more updated references for NPC (ref. 2 & 3). We increased additional comments to explain physiological relevance of our ALI culture model in lines from 154 to 163. We also addressed diagnostic value of examining susceptibility of nasopharyngeal epithelia to EBV infection in lines from 203 to 206. Application of our ALI infection model to develop anti-EBV therapy is also added in lines from 207 to 210.

Reviewer 2 Report

Yu et al. describe the generation of a model to study the role of EBV infection in the oncogenesis op nasoparyngeal cancer, which is a very relevant topic. As much as I appreciate the generation of a new model to study the ethiology of cancer, I worry about a couple of aspect of the manuscript.

Concerns;

  1. I don't understand the results described in section 2.2. It does not become clear to me what kind of experiments have been done and how it is shown that EBV infects this monolayer. Please revise this paragraph.
  2. EBV infection was only visualized by RNAish but was the amount of EBV enough/is it comparable to what is found in EBV+ tumors?
  3. I worry whether the way of transmission that has been used in this manuscript reflects the way transmission is done in humans. Please discuss.
  4. Have you tried the effect of EBV negative Akata cells on the monolayer? To what extend are your results based on adding the cells instead of EBV?
  5. The authors recognize that they have not shown the effect of EBV infection on cancer development. Thereby it is not clear whether the model can be used for EBV induced nasopharyngeal cancers.
  6. The authors propose a couple of experiments in the discussion, which should be done and added to this manuscript.

Author Response

 Concerns;

Q1) I don't understand the results described in section 2.2. It does not become clear to me what kind of experiments have been done and how it is shown that EBV infects this monolayer. Please revise this paragraph.

A1) We are sorry for shortage of explanation. To address readers who are not familiar with virological study, we have changed descriptions in section 2.2. We explained the method how EBV was transferred to epithelia. We also explained recombinant EBV with GFP marker. How EBV production was stimulated in latently infected Burkitt’s cells were also explained in detail (lines from 84 to 91).

Q2) EBV infection was only visualized by RNAish but was the amount of EBV enough/is it comparable to what is found in EBV+ tumors?

A2) We have stainned EBER in NPC tumor cells using conventional ISH method. However, we could observe weak signals. Therefore, we stained EBER using RNAscope to draw supplemental figure 2, which is attached to the revised manuscript. Moreover, we made explanation for the limitation of conventional EBER staining in lines 112 and 113 and in lines from161 to163.

Q3) I worry whether the way of transmission that has been used in this manuscript reflects the way transmission is done in humans. Please discuss.

A3) B lymphocytes are essential for establishing latent EBV infection. Persons with X-linked agammaglobulinemia who lack mature B lymphocytes are not infected with EBV ( Virol. 1999, 73, 1555-1564.). That means epithelial EBV infection is mostly mediated by B lymphocytes infected with EBV. We have explained physiological relevance of our assay in Discussion (lines from 154 to 163).

Q4) Have you tried the effect of EBV negative Akata cells on the monolayer? To what extent are your results based on adding the cells instead of EBV?

A4) We previously showed that engrafted NPC cells received EBV from virus-producing Akata B lymphocytes more efficiently (27.7%) than from cell-free virus (1.02%) (Otorhinolaryngol Head Neck Surg. 2018, 3, 1–7, Ref. 9). Though we did not make experiment using EBV-negative Akata cells, EBV-negative primary B lymphocytes were exposed to cell free EBV and immediately used for cell-mediated infection to epithelial cells (Shannon–Lowe C et al. 2006, 103, 7065-7070.). The efficiency of infection was increased by 103 to 104 fold compared to cell-free infection. We can expect similar enhancement in this model, because we have previously experienced more than 1,000-fold enhanced infection by cell-mediated infection (Nanbo A et al. J Gen Virol.2016, 97, 2989-3006, Ref. 13.).

Q5) The authors recognize that they have not shown the effect of EBV infection on cancer development. Thereby it is not clear whether the model can be used for EBV induced nasopharyngeal cancers.

A5) In vitro infection of EBV is known to transform primary B lymphocytes to lymphoblastic cell line. However, there is no report for transformation model on epithelial infection of EBV. Our observation that EBV infection disrupts epithelial integrity is very important. It is because the acquisition of local invasiveness is the first step to lead metastatic phenotype. Moreover, this model reminds us of cervical cancer caused by HPV infection, where HPV initially infects to basal stem cells then starts replication as the process of keratinocyte differentiation.

Q6) The authors propose a couple of experiments in the discussion, which should be done and added to this manuscript.

A6) We apologize for the vague discussion. The lack of a good transformation model in EBV-infected epithelial cells was the most problematic. We have fully considered the advantage of the current model. Our model has two contributions. Firstly, we showed that EBV was able to infect precancerous epithelial cells. Secondary, EBV infection decreased integrity of nasopharyngeal epithelial cells. Accordingly, more detailed discussions were described in lines from 197 to 206 in Discussion.

Round 2

Reviewer 2 Report

I thank the authors for their answer to my comments but actually they have not changed anything. It is still not clear to me for whom this models is useful as they have not shown EBV-related cancer development.

With a clear description what people can do with this models or to whom this paper will be informative the editor might reconsider but for now I dont see it, unforyunately.